# Diagnostic serology test comparison for Q fever and Rift Valley fever in humans and livestock from pastoral communities

**Valerie Hungerbühler**[1]*, **Ranya Özcelik**[1], **Mahamat Fayiz Abakar**[2], **Fatima Abdelrazak Zakaria**[2], **Martin Eiden**[3], **Sonja Hartnack**[4], **Pidou Kimala**[2], **Sonja Kittl**[5], **Janine Michel**[6], **Franziska Suter-Riniker**[7], **Salome Dürr**[1]

1 Veterinary Public Health Institute, Vetsuisse Faculty, University of Bern, Bern, Switzerland, 2 Institut de Recherche en Elevage pour le Développement, N'Djamena, Chad, 3 Institute for Novel and Emerging Infectious Diseases, Friedrich-Loeffler-Institut, Greifswald – Insel Riems, Germany, 4 Section of Epidemiology, Vetsuisse Faculty, University of Zurich, Zurich, Switzerland, 5 Institute of Veterinary Bacteriology, Vetsuisse Faculty, University of Bern, Bern, Switzerland, 6 Robert Koch Institute, Centre for Biological Threats and Special Pathogens, Highly Pathogenic Viruses, Berlin, Germany, 7 IFIK, Institute for Infectious Diseases, Faculty of Medicine, University of Bern, Bern, Switzerland

* valerie.hungerbuehler@unibe.ch

**Data Availability Statement:** All relevant data are within the manuscript and its Supporting Information files. All data are fully anonymized to

## Abstract

Q fever (QF) and Rift Valley fever (RVF) are endemic zoonotic diseases in African countries, causing significant health and economic burdens. Accurate prevalence estimates, crucial for disease control, rely on robust diagnostic tests. While enzyme-linked immunosorbent assays (ELISA) are not the gold standard, they offer rapid, cost-effective, and practical alternatives. However, varying results from different tests and laboratories can complicate comparing epidemiological studies. This study aimed to assess the agreement of test results for QF and RVF in humans and livestock across different laboratory conditions and, for humans, different types of diagnostic tests. We measured inter-laboratory agreement using concordance, Cohen's kappa, and prevalence and bias-adjusted kappa (PABAK) on 91 human and 102 livestock samples collected from rural regions in Chad. The serum aliquots were tested using ELISA in Chad, and indirect immunofluorescence assay (IFA) (for human QF and RVF) and ELISA (for livestock QF and RVF) in Switzerland and Germany. Additionally, we examined demographic factors influencing test agreement, including district, setting (village vs. camp), sex, age, and livestock species of the sampled individuals. The inter-laboratory agreement ranged from fair to moderate. For humans, QF concordance was 62.5%, Cohen's kappa was 0.31, RVF concordance was 81.1%, and Cohen's kappa was 0.52. For livestock, QF concordance was 92.3%, Cohen's kappa was 0.59, RVF concordance was 94.0%, and Cohen's kappa was 0.59. Multivariable analysis revealed that QF test agreement is significantly higher in younger humans and people living in villages compared to camps and tends to be higher in livestock from Danamadji compared to Yao, and in small ruminants compared to cattle. Additionally, RVF agreement was found to be higher in younger humans. Our findings emphasize the need to consider sample conditions, test performance, and influencing factors when conducting and interpreting epidemiological seroprevalence studies.

protect the privacy of the participants involved in the study.

**Funding:** This study was funded by the Swiss Federal Food Safety and Veterinary Office, grant number 1.17.09 (https://www.blv.admin.ch/blv/de/home.html), and the Wolfermann-Nägeli Foundation, grant number 2016/19 (https://stiftungen.stiftungschweiz.ch/organisation/wolfermann-naegeli-stiftung to S.D.). The funders had no role in study design, data collection and analysis, decision to publish, or preparation of the manuscript.

**Competing interests:** The authors have declared that no competing interests exist.

## Author summary

Q fever (QF) and Rift Valley fever (RVF) are zoonotic diseases that can be transmitted from animals to humans, causing health problems and economic losses in African countries. While various diagnostic tests for these diseases are available, they can be impractical, especially in resource-limited settings.

For this study, human and livestock samples from Chad were first tested in a local laboratory using a routine test (enzyme-linked immunosorbent assay). Serum aliquots were then sent to laboratories in Germany or Switzerland for retesting, using the same test type for livestock and a different test type for human samples.

We analysed the agreement between the test results and investigated the influence of the demographic characteristics of the sampled individual on this agreement. Our findings are crucial as they reveal discrepancies in test results, even though the samples originated from the same individuals. Additionally, we found that factors such as the age of the sampled individual influenced test agreement.

This study underscores the importance of considering sample conditions, test performance, and influencing factors when conducting and interpreting disease prevalence studies. Enhancing diagnostic procedures will aid in more effective disease control management, benefiting local communities and global health efforts.

## 1. Introduction

Q Fever (QF) and Rift Valley fever (RVF) are zoonotic diseases prevalent in several African countries [1]. Reported prevalence rates range from 7.8% to 39% for QF and 9.5% to 44.2% for RVF in livestock, and from 27% to 49.2% for QF and 13.2% to 28.4% for RVF in humans [1–4]. QF and RVF impact human health by causing a flu-like syndrome that can lead to a range of severe clinical manifestations. QF and RVF also result in significant production losses in animals due to abortions [5,6].

High-quality samples and robust diagnostic tests are essential for obtaining accurate prevalence estimates. Epidemiological studies play a critical role in generating the necessary data, subsequently influencing government prioritization of health interventions [7,8]. This prioritization is fundamental to effective disease control.

For QF diagnostics, the indirect immunofluorescence assay (IFA) can differentiate between acute and chronic infections and is regarded as the gold standard test for humans [9,10]. However, commercial IFA kits are not available for veterinary use [11]. The enzyme-linked immunosorbent assay (ELISA) is the most widely used test and is recommended by the WOAH (World Organisation for Animal Health) for rapid routine screening and large-scale epidemiological studies in ruminant populations [7]. For RVF, the virus neutralization test (VNT) is the most specific serological diagnostic test [8]. However, it requires living cell cultures, titered virus stocks, and highly trained personnel, making it labour-intensive, costly, and time-consuming (minimum 48 hours) [12]. Additionally, working with viable viruses poses a biohazard risk, making VNT unsuitable for use in laboratories without appropriate biosecurity facilities and vaccinated personnel [6,8]. For both diseases, ELISAs offer a rapid, cost-effective, and practical alternative with less stringent biosafety requirements, making them suitable for routine use in low- and middle-income countries (LMIC) [7,8].

The use of ELISA in some studies and the gold standard test, which may differ between countries, in others can lead to discrepancies in estimated prevalence, making comparisons challenging; thus, harmonized monitoring and reporting schemes for QF and RVF have been proposed to enable consistent comparisons over time and across countries [13–15].

Several studies have assessed the inter-test agreement of ELISA for QF and RVF compared to other diagnostic tests, reporting variable agreement ranging from poor to good for QF and from good to excellent for RVF [16–22]. Diagnostic test validation can be achieved through various methods, including assessing the agreement between different tests without assuming one as the gold standard [23]. Concordance, the proportion of test results in agreement over the number of all tests performed, is a straightforward measure but does not account for agreement beyond chance. Therefore, Cohen's kappa statistic, which adjusts for random matches, is often used to measure the agreement between two test results [23]. Cohen's kappa values range from zero (agreement is equal to that expected by chance) to one (complete agreement beyond chance), with benchmarks between agreement categories varying among authors [24–26]. Although Cohen's kappa is a standard measure, it has limitations such as prevalence and bias effects. Prevalence effects arise when the proportion of positive results deviates significantly from 50% [27]. The effect of prevalence depends on the method of modeling agreement and can substantially reduce kappa values [27]. Bias effects occur when there is a disparity in the proportion of positive results between the two tests, which leads to reduced kappa values [27]. To address these effects, the prevalence- and bias-adjusted kappa (PABAK) can be calculated [28,29].

The reasons for disagreement between diagnostic tests have rarely been thoroughly investigated. Potential factors include poor sample quality, variability in tests used, and discrepancies arising from the same test being conducted in different laboratories [18,22,30]. Additionally, biological factors such as age, sex, other diseases, and species may influence the consistency of test results for the same sample. Previous studies have suggested associations between test performance and variables such as region, age, and livestock species [31–33]. However, these studies have not provided conclusive evidence or statistical significance.

The objective of this study was to assess the inter-laboratory agreement of standard diagnostic tests utilized by the participating laboratories, measured by the concordance, Cohen's kappa, and PABAK. The comparison involved results obtained from commercial ELISA tests conducted in a laboratory in Chad and results obtained from ELISA and indirect IFA tests performed on livestock and human serum aliquots, respectively, in laboratories in Germany and Switzerland. Additionally, we evaluated the influence of demographic factors on the agreement between the two test results. The study enhances our understanding of the inter-laboratory agreement of diagnostic test results across laboratory conditions and, for humans, test types, which is crucial for accurately interpreting results from epidemiological seroprevalence studies.

## 2. Material and Methods

### 2.1. Ethics statement

The study was approved by the Ethics Committee of Northwest and Central Switzerland (EKNZ) (project id 2017–00884) and by the Comité National de Bioéthique du Tchad (CNB-Tchad) (project id 134/PR/MESRS/CNBT/2018). Formal written consent was obtained from study participants and animal owners after we presented our study to the community and before data collection occurred.

## 2.2. Sample collection and laboratory analysis in Chad

The samples analysed in this study were collected between January and February 2018, as reported by Özcelik, R., et al [4]. In brief, a cross-sectional study in livestock (cattle, sheep, goats, horses, and donkeys) and human populations was conducted in the two rural health districts, Yao and Danamadji, in Chad. Multistage cluster sampling was used, with villages and nomadic camps serving as cluster units. The clusters were chosen based on human population size. The sample size was calculated using R software, assuming a 50% prevalence for Q fever, and RVF, with a 95% confidence interval. The sample size for two-stage cluster sampling was determined using a formula incorporating the sample size for simple random sampling, the intra-cluster correlation coefficient of 0.2, and the number of 20 individuals sampled per cluster [4]. In Danamadji and Yao, respectively, blood samples were collected from 571 and 389 humans and 560 and 483 livestock. No individuals showing clear signs of illness were sampled, and similarly, animals that were visibly ill and hence often isolated were also excluded from the sampling [4].

The samples were subsequently analysed at the Institut de Recherche en Élevage pour le Développement (IRED) in N'Djamena, Chad. Livestock and human samples were analysed using different indirect ELISAs: ID Screen Q Fever Indirect Multi-species ELISA from IDvet for livestock and the Panbio Coxiella burnetii IgG ELISA from Abbott for humans. For RVF, a competitive ELISA (ID Screen Rift Valley Fever Competition Multi-species from IDvet) was used for human and livestock samples. The diagnostic test procedure and thresholds were applied according to the manufacturer's protocols without modification (S1 Table). Equivocal samples were retested once.

## 2.3. Diagnostic testing in Switzerland and Germany

Following the initial diagnostic analysis at IRED, 10% of the human and livestock samples from each region were randomly selected and sent as serum aliquots to laboratories in Switzerland and Germany in 2021 for repeated diagnostic analysis for QF and RVF, respectively. The decision to use a 10% subsample of the original sample size was based on the resources available. In Switzerland, two indirect ELISAs (IDEXX Q Fever IgG Antibody and ID Screen Q Fever Indirect Multispecies from IDvet) were used at the Center for Zoonoses, Animal Bacterial Diseases, and Antimicrobial Resistance (ZOBA) for QF diagnostics in livestock samples (ruminants and equids, respectively). At the Institute for Infectious Diseases (IFIK) of the University of Bern, an indirect IFA (Q Fever IFA IgG assay from Focus Diagnostics, US) was used for QF diagnostics in human samples. For RVF, livestock samples were analysed using a competitive ELISA (ID Screen Rift Valley Fever Competition Multispecies ELISA from IDvet) at the Friedrich-Loeffler-Institute (FLI), and human samples were analysed using an indirect IFA (Anti-Rift-Tal-Fieber-Viren-IIFT [IgG] from EUROIMMUN) at the Robert Koch Institute (RKI). The diagnostic test procedure and thresholds were applied according to the manufacturer's protocols without modification (S1 Table). Equivocal samples were not retested.

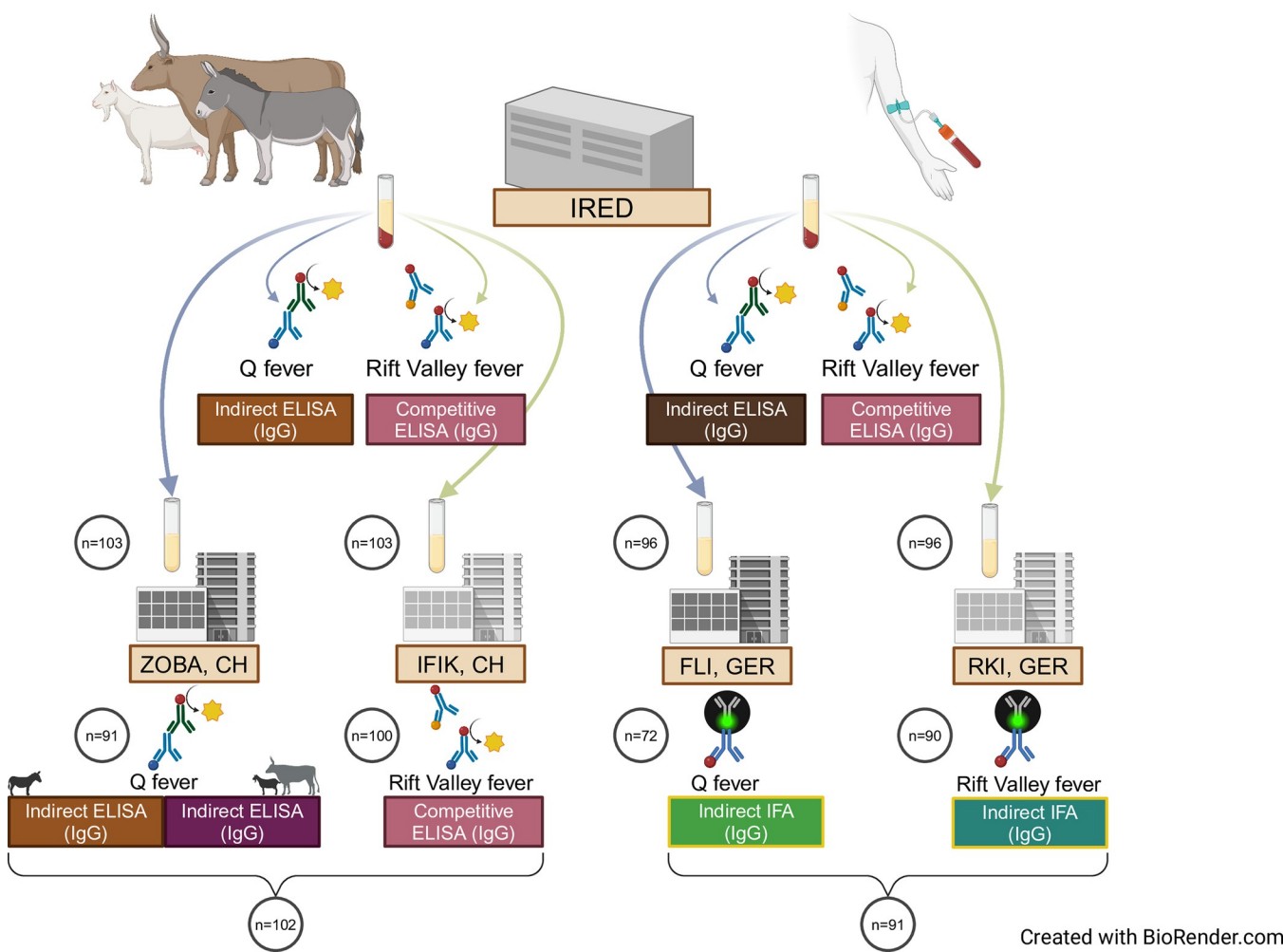

**Fig 1. Sample collection and laboratory analysis were conducted at the Institut de Recherche en Élevage pour le Développement (IRED) in Chad, with subsequent diagnostic testing performed in Switzerland (CH) and Germany (GER).** The specific tests used in each laboratory are noted, along with the antibodies detected (IgG = Immunoglobulin G). The number of samples processed at each stage is provided. Different colors in the diagnostic tests indicate that assays from different manufacturers were employed. Of the original 103 livestock and 96 human samples sent to Germany and Switzerland, 102 livestock and 91 human samples underwent either both QF and RVF diagnostics or one of the tests. One livestock sample and five human samples were unavailable for any diagnostic procedure and were fully excluded from further analysis. Abbreviations: ZOBA: Center for Zoonoses, Animal Bacterial Diseases, and Antimicrobial Resistance; IFIK: Institute for Infectious Diseases; FLI: Friedrich-Loeffler-Institute; RKI: Robert Koch Institute; ELISA: Enzyme-linked Immunosorbent assay; IFA: Immunofluorescence assay. Figure created with BioRender.com.

## 2.4. Sample processing

Of the 206 serum aliquots from 103 livestock samples and 192 serum aliquots from 96 human samples sent to laboratories in Switzerland and Germany, 15 livestock aliquots and 30 human aliquots were excluded from further analysis (Table 1). Reasons for exclusion included unsuccessful matching of test identities due to labelling errors, missing serum upon arrival, and equivocal results (Table 1). As a result, one livestock sample and five human samples were fully excluded from the study sample. Ultimately, 91 human and 102 livestock samples were tested using one or both assays and were included in the agreement testing (Table 1, Fig 1).

**Table 1. Overview of diagnostic tests from Chad and Switzerland (CH) or Germany (GER).**

| Disease | QF[1] | | RVF[2] | |
|---|---|---|---|---|
| Samples | Livestock | Humans | Livestock | Humans |
| Laboratory, Chad | IRED[3] | IRED[3] | IRED[3] | IRED[3] |
| Test 1, Chad | **IDvet** ID Screen Q Fever Indirect Multi-Species | **Abbott** Panbio Coxiella burnetii IgG ELISA | **IDvet** ID Screen Rift Valley Fever Competition Multi-species | **IDvet** ID Screen Rift Valley Fever Competition Multi-species |
| Laboratory, CH/GER | ZOBA (CH)[4] | IFIK (CH)[5] | FLI (GER)[6] | RKI (GER)[7] |
| Test 2, CH/GER | **IDEXX** Q Fever Ab Test (ruminants) **IDvet** ID Screen Q Fever Indirect Multispecies (equids) | **Focus Diagnostics** Q Fever IFA IgG assay | **IDvet** ID Screen Rift Valley Fever Competition Multi-species | **EUROIMMUN** RVF IFA IgG assay |
| Initial number of serum aliquots sent to CH/GER | 103 | 96 | 103 | 96 |
| Excluded serum aliquots from test 2 due to missing serum | 2 | 8 | 0 | 4 |
| Equivocal results during test 2 | 8 | 14 | 2 | 0 |
| Non-matchable | 2 | 2 | 1 | 2 |
| Number of serum aliquots used for agreement testing | **91** | **72** | **100** | **90** |

[1]Q fever

[2]Rift Valley fever

[3]Institut de Recherche en Elevage pour le Développement, Chad

[4]Center for Zoonoses, Animal Bacterial Diseases and Antimicrobial, Switzerland (CH)

[5]Institute for Infectious Diseases, Switzerland (CH)

[6]Friedrich-Loeffler-Institute, Germany (GER)

[7]Robert Koch Institute, Germany (GER)

## 2.5. Statistical analysis

The inter-laboratory agreement of the test results from Chad and Switzerland or Germany was evaluated using concordance, Cohen's Kappa, and PABAK for each of the four datasets: QF results in human samples, QF results in livestock samples, RVF results in human samples, and RVF results in livestock samples. Cohen's kappa and PABAK values were interpreted according to the standard scale: 'fair' agreement (kappa = 0.21–0.40), 'moderate' agreement (kappa = 0.41–0.60), 'substantial' agreement (kappa = 0.61–0.80), and 'almost perfect' agreement (kappa > 0.80) [26].

In addition, we investigated factors associated with test agreement by assigning to each sample a value of 0 if there was disagreement between the two test results (i.e., negative in Chad and positive in Switzerland/Germany, or the opposite) and 1 if the test results were consistent (i.e., both positive or both negative). This binary outcome was used as the dependent variable in logistic regression models to identify the statistical association between test agreement and demographic factors, including the district (Yao *versus* Danamadji) and setting (village *versus* camp) where the sample was collected, and sex, age, and livestock species (cattle, small ruminants, and equids) of the sampled individual. The variable age was analysed in two ways, as a continuous and as a categorical variable. For the continuous variable, a unit of 1 year was used for livestock and of 10 years for humans. For the categorical variable, samples were stratified as < 2 years (age group 1), 2–3 years (age group 2), 4 years and older (age group 3) for livestock, and < 30 years (age group 1), 30–39 years (age group 2), 40–60 years (age group 3), 61 years and older (age group 4) for humans.

Univariable logistic regressions were initially performed to assess individual predictors. In cases where the univariable model was infeasible due to perfect agreement in one group, Chi-square tests were applied. Odds ratios (OR) and their corresponding 95% confidence intervals were calculated for these analyses. To consider potential interdependencies between the variables, we applied multivariable logistic regressions to estimate adjusted coefficients and OR. We included all variables and selected age as categorical variable.

Statistical calculations, modeling, and data visualization were conducted in R (version 4.2.2). The package "irr" was used to calculate the concordance and Cohen's kappa. The package "vcd" was used to obtain confidence intervals of Cohen's kappa, computed using the standard method based on normal approximation [34]. The epi.kappa() function from the "epiR" package was used to calculate PABAK and corresponding confidence intervals. The confidence intervals for the OR were calculated using the output values of the associated regression model, with the upper and lower CIs derived using the function exp(confint.default(model)).

## 3. Results

### 3.1. Sample population

Of the 91 human and 102 livestock samples that were used to perform the inter-laboratory test agreement analysis, most (62% of human and 57% of livestock) samples were collected from Danamadji (S2 Table). Fifty-six percent of human samples and 58% of livestock samples were collected from camps. The sex distribution was uneven, with 70% of human sampled being male and 70% of livestock sampled being females. Among humans, age groups 1–3 were evenly represented (30%, 28%, 33%), while only 9% belonged to age group 4. In livestock, 50% of the samples were from age group 2, with 17% and 33% from age groups 1 and 3, respectively. Most livestock samples were from cattle (46%), followed by small ruminants (41%) and equids (13%) (S2 Table). The livestock sampled originated from 24 different herds (one herd per cluster).

### 3.2. Diagnostic tests agreement

**3.2.1 Level of inter-laboratory test agreement.** Concordance values ranged from 62.5% to 94% (Table 2). Cohen's kappa values, which ranged from 0.31 to 0.59, indicated that livestock QF and RVF, and human RVF tests had 'moderate' agreement, while human QF tests had 'fair' agreement (Table 2). PABAK values showed that the livestock QF and RVF tests had 'almost perfect' agreement, the human RVF test had 'substantial' agreement, and the human QF tests had 'fair' agreement (Table 2).

**Table 2. Analyses of test agreement (concordance and Cohen's kappa) of samples tested in an inter-laboratory test agreement study.**

| | Switzerland | | | | Germany | | | |
|---|---|---|---|---|---|---|---|---|
| | QF[1] livestock | | QF[1] humans | | RVF[2] livestock | | RVF[2] humans | |
| Chad | Pos | Neg | Pos | Neg | Pos | Neg | Pos | Neg |
| Pos | 6 | 3 | 22 | 3 | 5 | 4 | 16 | 9 |
| neg | 4 | 78 | 24 | 23 | 2 | 89 | 8 | 57 |
| Concordance (%) | 92.3 | | 62.5 | | 94.0 | | 81.1 | |
| Cohen's kappa value (95% CI) | 0.59 (0.31, 0.86) | | 0.31 (0.13, 0.49) | | 0.59 (0.30, 0.89) | | 0.52 (0.33, 0.72) | |
| *PABAK* value (95% CI) | 0.846 (0.696, 0.937) | | 0.25 (0.006, 0.473) | | 0.88 (0.748, 0.955) | | 0.622 (0.43, 0.771) | |

[1]Q fever

[2]Rift Valley fever

**3.2.2. Influence of factors on inter-laboratory test agreement.** For QF in livestock, none of the investigated demographic factors significantly impacted the agreement between the two test results in both univariable and multivariable analyses (Tables S3 and 3). However, some notable trends (p < 0.15) emerged: small ruminants tended to show better agreement than cattle, and samples from Yao showed lower agreement compared to those from Danamadji (Table 3). For QF in humans, samples from villages had significantly higher agreement compared to those from camps, with odds of agreement being 13.4 times higher (Table 3). Additionally, older age groups had significantly lower agreement compared to the youngest age group (Table 3).

**Table 3. Results of multivariable logistic regression models investigating the effect of demographic factors as independent variables on the inter-laboratory test agreement of Q fever (QF) and Rift Valley fever (RVF) in humans and livestock.** Goodness of fit of the models are presented as pseudo-R-squared ($R^2$). Bold p-values indicate significance based on a threshold of 0.05.

| | Switzerland | | Germany | |
| --- | --- | --- | --- | --- |
| | QF[1] livestock | QF[1] humans | RVF[2] livestock | RVF[2] humans |
| Independent variable (OR and 95% CI) P-value (P) | | | | |
| District (Reference: Danamadji) | Yao: 0.14 (0.01, 1.74) P = 0.13 | Yao: 1.38 (0.38, 4.99) P = 0.63 | Yao: 4.99 (0.41, 60.49) P = 0.21 | Yao: 1.37 (0.37, 5.04) P = 0.64 |
| Setting (Reference: camp) | Village: 0.34 (0.05, 2.06) P = 0.24 | Village: 13.35 (3.20, 55.86) **P = 0.0004** | Village: 0.32 (0.04, 2.43) P = 0.27 | Village: 1.85 (0.58, 5.93) P = 0.30 |
| Species (Reference: cattle) | Equids: 5.51 (0.29, 103.21) P = 0.25 Small ruminants: 6.26 (0.59, 65.76) P = 0.13 | - | Equids: Inf (perfect agreement) Small ruminants: 0.65 (0.09, 4.78) P = 0.67 | - |
| Age group (Reference: age group 1/ group 2*) | Group 2: 1.30 (0.10, 17.39) P = 0.84 Group 3: 0.93 (0.06, 14.38) P = 0.96 | Group 2: 0.67 (0.15, 3.07) P = 0.61 Group 3: 0.14 (0.03, 0.82) **P = 0.03** Group 4: 0.09 (0.01, 0.83) **P = 0.03** | Group 1*: Inf (perfect agreement) Group 3*: 0.30 (0.04, 2.41) P = 0.26 | Group 2: 0.62 (0.12, 3.16) P = 0.57 Group 3: 0.32 (0.07, 1.47) P = 0.14 Group 4: 0.16 (0.02, 1.14) P = 0.07 |
| Sex (Reference: male) | Female: 0.38 (0.03, 4.61) P = 0.45 | Female: 0.82 (0.20, 3.38) P = 0.78 | Female: 0 (perfect agreement in males) | Female: 1.01 (0.27, 3.86) P = 0.99 |
| $R^2$ | 0.23 | 0.35 | 0.28 | 0.09 |

[1] Q fever

[2] Rift Valley fever

* In the multivariable logistic regression model, age group 2 was used as the reference for both age group 1 and age group 2 for RVF livestock, due to perfect agreement observed in age group 1.

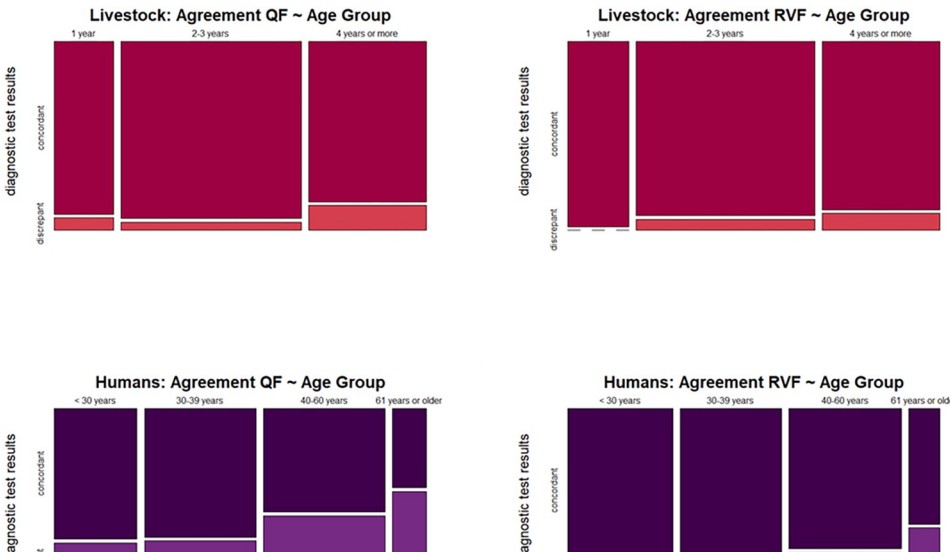

**Fig 2. Mosaic plot illustrating the association between age groups and inter-laboratory test agreement, the widths of the adjacent rectangles represent the proportion of individuals within each age group.** The heights of the rectangles represent the proportion of individuals within each age group who exhibit a specific test agreement outcome (concordant or discrepant). Abbreviations: QF: Q fever; RVF: Rift Valley fever.

For RVF in humans and livestock, none of the demographic factors significantly influenced test agreement. However, some trends were observed for RVF test agreement in humans, with older age groups showing lower agreement compared to the youngest age group (Table 3).

Across diseases and populations, there was a consistent trend of lower agreement with increasing age, which was significant for human QF and almost reached significance for human RVF (Table 3 and Fig 2). For livestock RVF tests, the odds ratio for agreement was also lower in older age groups, although the p-value was 0.26 (Table 3).

## 4. Discussion

Our study revealed varying levels of test agreement, ranging from fair to moderate (Cohen's kappa) or almost perfect when considering PABAK. The good inter-laboratory agreement of livestock test results for RVF was in line with other studies on RVF test agreement, although it is important to note that literature on this subject is limited [20,21]. Similarly, the body of research on test agreement for livestock QF is also limited. The inter-laboratory agreement for livestock QF observed in our study was slightly lower than the values reported in previous studies that evaluated agreement between different ELISA tests or between ELISA and IFA [18,19]. However, those previously reported test comparisons were performed within the same laboratories and time frame, which may explain the observed differences in agreement. Notably, the livestock RVF agreement was assessed using the same ELISA from the same manufacturer in both Chad and Germany. In contrast, for livestock QF, only the equid samples were tested using the same ELISA in Switzerland as in Chad, whereas ruminant samples were analyzed with an ELISA from a different manufacturer. The latter ELISA was adapted for automation in ruminants and was therefore the standard assay used by the participating laboratory for this species. This discrepancy may have contributed to the better inter-laboratory agreement observed for livestock RVF compared to livestock QF.

We observed a notably lower agreement for human tests compared to livestock tests for both diseases, which may be attributed to using two different diagnostic methods for human samples, ELISA in Chad and indirect IFA in Switzerland or Germany. Previous studies have shown varying sensitivities and specificities of the commercial diagnostic test for QF used in this study (Panbio from Abbott), ranging from 71% to 100% [10,35–37]. In these studies, indirect IFA was used as a reference method to evaluate the ELISA, revealing varying agreement between the results of the two tests. The variability in sensitivity for the same test raises the question of whether it is due to the scope for interpretation in indirect IFA, which is considered a challenge, even though indirect IFA is regarded as the gold standard for human QF diagnostics [5,7]. A study that compared QF indirect IFA results from different reference centres in three countries (United Kingdom, France, and Australia) found a concordance between the indirect IFA results of only 35% [38]. Our results reflect this uncertainty and underline the complexity of QF diagnostics [5,7,39]. A previous study comparing IFA (the same test used in our study) and ELISA (from a different manufacturer than the one used in our study) found that IgG phase II antibody titers decline more slowly when measured by IFA, with significantly higher detection rates after one year [40]. These findings indicate that IFA may offer superior sensitivity and longer-lasting detection of IgG phase II antibodies compared to ELISA, which may account for the lower agreement observed in the human QF tests in our study. Similarly, for RVF, the lower inter-laboratory agreement for human tests compared to livestock tests may be attributed to the use of different testing methods in Chad (ELISA) and Germany (indirect IFA). The reported sensitivity and specificity values for the ELISA kit used in Chad are 98% and 100%, respectively [20]. While the IFA test used in Germany (EUROIMMUN RVF IFA IgG assay) is a certified assay, its performance has not been evaluated by external independent assessment [12]. The existing literature on RVF diagnostic tests, particularly IFA, is limited. Generally, IFA can achieve high specificity and sensitivity; however, these metrics are highly dependent on the operator's skill in interpreting fluorescent signals under a microscope and the quality of the reagents used [41].

Although quantitative information on the quality of the individual samples in our study are unavailable, laboratory staff in Switzerland and Germany reported concerns regarding haemolysis. Given that haemolysis is a well-documented cause of test result discrepancies in clinical laboratories [42,43], it may have contributed to the observed disagreement between test results in our study. Although serum was sent to the laboratories in Germany and Switzerland, haemolysis may have occurred before or during the process of separating serum from whole blood. Consequently, the serum could contain haemoglobin and other intracellular components from lysed red blood cells, potentially compromising the quality of the sample and interfering with laboratory tests [43]. There are many in vitro causes of haemolysis, mostly pre-analytical problems such as incorrect procedures and/or materials used in blood collection, while transport, processing, and storage account for only a minority of cases [44]. This limitation shows the importance of careful planning and execution of the pre-analytical phase, especially in prevalence studies where the outcome can be influenced by the quality of the sample material. Nevertheless, it is crucial to recognize the challenges associated with collecting samples under difficult field conditions, where access to centrifuges may be limited until several days after blood sampling. In addition, the samples in our study were stored for 2.5 years with repeated freeze-thaw cycles between the performance of the two tests for some of the samples, which probably affected sample quality. Several studies have assessed the effects of repeated freeze-thaw cycles on IgG stability, although not specifically on Anti-QF/RVF IgG, and found minimal impact [45,46]. However, the effects of prolonged storage have not been explored. Future research should investigate whether extended storage, such as the period between processing at the local

laboratory and further analysis at secondary laboratories, could compromise sample quality, and consequently test results. We emphasize the importance of considering these challenges when discussing the outcomes of epidemiological prevalence studies or diagnostic test evaluation studies.

Our results demonstrate a statistical relationship between test agreement and age of the sampled individuals, with higher agreement observed in younger individuals for both diseases and in humans and livestock. This finding aligns with a study by de Bronsvoort et al. (2019), which suggested that lower agreement in older individuals may be due to their higher likelihood of previous exposure to other pathogens over their lifetime [31]. This may result in cross-reactivity in serological tests, making serological differentiation between diseases more difficult [39]. Cross-reactions caused by antibodies induced by other pathogens, such as *C. burnetii* antigens, the agent causing QF, with antibodies produced against *Bartonella spp.*, *Legionella spp.*, and *Chlamydiae spp.* has been reported [47–49], or RVF virus antigens with antibodies produced against Rio Grande virus [50–52]. Furthermore, with increasing age, the likelihood of exposure to the causative agent of QF and RVF increases [53,54], as does the chance of having residual antibodies against these diseases in the blood [6,55–57]. In addition, in QF, antigen-lipopolysaccharide complexes remaining in the host after infection with *C. burnetii* can trigger humoral and cell-mediated immune responses, producing interfering antibodies [58,59]. Such antibody titres can potentially produce ambiguous results that are neither clearly positive nor negative, leading to misinterpretation and discordant test results. Additionally, immunosenescence–the gradual deterioration of the immune system associated with aging–may also play a role in the increased disagreement of diagnostic test results among older individuals. Immunosenescence is characterized by reduced immune function, altered antibody responses, and an increased tendency toward chronic inflammation, including the presence of proinflammatory "age-associated B cells," which have been shown to produce autoantibodies [60–62]. These autoantibodies may interfere with antigens used in diagnostic tests; a hypothesis that warrants further investigation. Consequently, these findings underscore the need for further research to develop age-specific diagnostic protocols, such as adjusting cut-off values for interpreting results, and to improve test accuracy across diverse populations.

The district-level analysis showed a trend toward higher agreement for livestock QF tests in samples from Danamadji compared to Yao. This regional variation may be influenced by varying local environmental conditions, disease prevalence, or livestock management practices. However, we did not capture such information, so we were not able to identify a potential latent factor that can explain the difference in test agreement detected for the two regions. Continuing with the focus on sampling location, the sample collection setting (village versus camp) was identified as a significant factor in the multivariable model for human QF test agreement. One hypothesis for this finding is that villages generally have shorter distances from the sampling location to the laboratory, allowing for more appropriate storage conditions compared to remote camps. This shorter travel time likely preserves sample integrity, resulting in higher test agreement. The significant role of the setting variable suggests that environmental and logistical factors may be crucial for diagnostic test result interpretation, although we did not observe the influence of the setting for RVF in humans nor for livestock species. Future studies should investigate the impact of environmental and logistical factors on test results and gather detailed data on local conditions and practices that affect sample quality and test agreement.

Small ruminants exhibited a trend towards higher agreement in QF test results compared to cattle. In a different type of QF ELISA, Stellfeld et al. (2020) observed significant differences in the ranges of $OD_{450}$ values obtained from sera of sheep, goats, and cattle [63]. Sheep exhibited a larger range of $OD_{450}$ values, whereas cattle showed a smaller range [63]. Greater

distribution between $OD_{450}$ values allows for more precise grading of ELISA results, which may explain the better inter-test agreement for small ruminants compared to cattle. These differences in $OD_{450}$ ranges are likely due to species-specific immune responses to *C. burnetii*, suggesting varying immune reactions among ruminant species [63].

Although females showed lower inter-laboratory test agreement compared to males for both diseases and populations, the high p-values and wide confidence intervals of the ORs in our data did not indicate any significant influence of sex on the inter-laboratory test agreement.

## Conclusion

Our study highlights the variability in inter-laboratory diagnostic test agreement for QF and RVF serology in humans and livestock based on samples collected in Chad. Despite differences in laboratories, personnel, and test types, test agreements ranged from fair to moderate (Cohen's kappa) or almost perfect considering PABAK. Given the reliance on serological profiles for QF and RVF epidemiological studies, it is crucial to consider factors that may complicate accurate diagnosis. We identified that human QF test agreement was significantly higher in individuals living in villages and younger individuals, with the latter trend also observed in human RVF tests. Our findings emphasize the need to recognize that diagnostic tests may yield varying results, impacting the outcome and interpretation of disease prevalence studies. If resources permit, it is recommended to confirm positive results by retesting with the gold standard test.

## Supporting information

**S1 Table. This document (S1A–S1F Tables) provides a summary of the test protocols for all diagnostic assays employed in this study.**
(DOCX)

**S2 Table. This table presents the demographic characteristics of the sample population.**
(DOCX)

**S3 Table. This table presents the findings from the univariable logistic regression models.**
(DOCX)

**S1 Dataset. This dataset consists of the data required to replicate all study findings reported in the article for the human population.**
(XLSX)

**S2 Dataset. This dataset consists of the data required to replicate all study findings reported in the article for the livestock population.**
(XLSX)

## Declaration of generative AI and AI-assisted technologies in the writing process

While preparing this work, the author used ChatGPT to correct the English of revised sentences (using the command: 'correct the English'). After using this tool, the author reviewed and edited the content as needed and takes full responsibility for the content of the publication.

## Author Contributions

**Conceptualization:** Valerie Hungerbühler, Ranya Özcelik, Mahamat Fayiz Abakar, Sonja Hartnack, Salome Dürr.

**Data curation:** Valerie Hungerbühler.

**Formal analysis:** Valerie Hungerbühler.

**Funding acquisition:** Salome Dürr.

**Investigation:** Ranya Özcelik, Mahamat Fayiz Abakar, Fatima Abdelrazak Zakaria, Martin Eiden, Pidou Kimala, Sonja Kittl, Janine Michel, Franziska Suter-Riniker, Salome Dürr.

**Project administration:** Valerie Hungerbühler, Ranya Özcelik, Mahamat Fayiz Abakar, Salome Dürr.

**Resources:** Mahamat Fayiz Abakar, Fatima Abdelrazak Zakaria, Martin Eiden, Sonja Hartnack, Pidou Kimala, Sonja Kittl, Janine Michel, Franziska Suter-Riniker, Salome Dürr.

**Supervision:** Salome Dürr.

**Validation:** Salome Dürr.

**Visualization:** Valerie Hungerbühler.

**Writing – original draft:** Valerie Hungerbühler.

**Writing – review & editing:** Valerie Hungerbühler, Salome Dürr.

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
