## [Decision Letter · Decision Letter 0]

19 Aug 2024

Dear DVM Hungerbühler,

Thank you very much for submitting your manuscript "Diagnostic serology test comparison for Q fever and Rift Valley fever in humans and livestock from pastoral communities" for consideration at PLOS Neglected Tropical Diseases. As with all papers reviewed by the journal, your manuscript was reviewed by members of the editorial board and by several independent reviewers. The reviewers appreciated the attention to an important topic. Based on the reviews, we are likely to accept this manuscript for publication, providing that you modify the manuscript according to the review recommendations. 

Reviewer #2: Manuscript with reviewer's comments attached

Sincerely,

Philip Mshelbwala

Academic Editor

Elvina Viennet

Section Editor

Reviewer #2: Manuscript with reviewer's comments attached

Reviewer's Responses to Questions

**Key Review Criteria Required for Acceptance?**

**Methods**

-Are the objectives of the study clearly articulated with a clear testable hypothesis stated?

-Is the study design appropriate to address the stated objectives?

-Is the population clearly described and appropriate for the hypothesis being tested?

-Is the sample size sufficient to ensure adequate power to address the hypothesis being tested?

-Were correct statistical analysis used to support conclusions?

-Are there concerns about ethical or regulatory requirements being met?

Reviewer #1: -Are the objectives of the study clearly articulated with a clear testable hypothesis stated? 

Yes

-Is the study design appropriate to address the stated objectives?

Not entirely. My concerns about this are stated in the "Summary and General Comments" box below

-Is the population clearly described and appropriate for the hypothesis being tested?

Yes

-Is the sample size sufficient to ensure adequate power to address the hypothesis being tested?

There was no sample size justification or posthoc power analyses

-Were correct statistical analysis used to support conclusions?

Yes

-Are there concerns about ethical or regulatory requirements being met?

No

Reviewer #2: The objectives of the study are very clear and the study design was appropriate. The researchers gave a clear description of their population, sampling technique, and used relevant statistical analysis. Standard ethical approval was obtained as evidenced by the name of the ethics committee and the approval number. Details of the sample size determination was however not included.

Reviewer #3: The authors evaluated the agreement between a laboratory in Chad and laboratories in Germany and Switzerland. This study is interesting because it sheds light on the reliability of diagnostic tests performed in developing countries, despite the lack of adequate biosecurity facilities. The sampling, sample size, and statistical analysis are sufficient to support the conclusions.

Reviewer #4: The paper aims to investigate the differences between diagnostic tests of Q fever and RVF in livestock and human in different labs. The study design and the statistical analysis would support the testing of the comparison. However, the sampling design was not clearly introduced. The main concerns are: 1. need more demographic information of livestock from the two areas: by species: age, sex, herd sizes, and all livestock from one herd? 2.for the clustering sampling: within each cluster how to choose the apparently healthy ones? how did the authors determine the sample size then?

**Results**

-Does the analysis presented match the analysis plan?

-Are the results clearly and completely presented?

-Are the figures (Tables, Images) of sufficient quality for clarity?

Reviewer #1: -Does the analysis presented match the analysis plan?

Not entirely. My concerns about this are stated in the "Summary and General Comments" box below

-Are the results clearly and completely presented?

Yes

-Are the figures (Tables, Images) of sufficient quality for clarity?

Not entirely. My concerns about this are stated in the summary box below

Reviewer #2: The results were presented in a clear cut manner. Appropriate data visualizations were used to effectively communicate the research findings. Both presented and supplementary figures were sufficient.

Reviewer #3: (No Response)

Reviewer #4: The results are well presented in general. However, a few places could be improved:

1. lines 202-207: this paragraph should be included in the method section

2. Fig 1: Please add the testing methods, institutes and countries aside the buildings so easier for readers to understand

3. line 222: suggest revising "men" to "males" as the studying group of <30 may had included boys 

4. Table 3: the table looks incomplete at the bottom? and the "*" needs to explained in a note under.

**Conclusions**

-Are the conclusions supported by the data presented?

-Are the limitations of analysis clearly described?

-Do the authors discuss how these data can be helpful to advance our understanding of the topic under study?

-Is public health relevance addressed?

Reviewer #1: -Are the conclusions supported by the data presented?

Yes

-Are the limitations of analysis clearly described?

Not entirely

-Do the authors discuss how these data can be helpful to advance our understanding of the topic under study?

Yes

-Is public health relevance addressed?

Indirectly; seroprevalence studies provide data for making policies. This study highlights concerns that need to be considered/addressed whenever using seroprevalence data

Reviewer #2: The authors' conclusion was comprehensible and in accordance with the presented data. In addition, their findings were competently compared with current scientific reports. The study limitations and factors likely to be responsible for the disparities observed in the study were highlighted and areas for further research were indicated. The authors elucidated the public health importance and epidemiological relevance of their study.

Reviewer #3: The conclusion is superficial and requires more detail, as indicated in the attached file.

Reviewer #4: In general, the conclusions were well presented. However, the limitation of the study could have been addressed better by discussing why not using the same methods in different labs to compare. For example, authors mentioned the poor consistent may be due to different tests used in different labs, but why can't use the same IFA in Chad to compare? Need to discuss the challenge for having other tests than ELISA in Chad. 

Other comments:

1. Lines 270-273: No need to repeat this as it has been introduced.

2. Line 291: please elaborate on the possible misinterpretations

3. Line 321: Since you mentioned the age had impacts on antibody titres, please advise on the implementation of this finding. For example, what if you adjust the cut-offs of reading results?

**Editorial and Data Presentation Modifications?**

Reviewer #1: Minor Revision

Reviewer #2: Minor editorial modifications required have been highlighted in the reviewed manuscript attached to this submission. The authors should include the details of sample size determination. This information is crucial for research reproducibility.

Reviewer #3: I have attached a file detailing suggested changes and indicating areas that need more details. I recommend minor revisions.

Reviewer #4: Minor Revision

**Summary and General Comments**

Reviewer #1: This study sought to address an often-overlooked concern with seroprevalence studies. The sampling technique used, the diagnostic performance of the laboratory test, the diagnostic quality of the sample type, and the lab-to-lab variation in out-of-kit reagents, personnel, and SOPs are important factors that drive the validity of findings from such studies. Specifically, this study evaluates the agreement of Q fever and RVF serological results in a laboratory in Chad compared to results from laboratories in 2 European countries. The findings of this study are most relevant to epidemiologists.

Overall, the authors were clearly able to justify the study and discuss the results in-depth. I have a few concerns:

Major concerns: 

Line 138: I missed the portion of the methods that explains how they came about the sample size; it would be great if the authors included some sample size justification, if there was one, or indicate if it was non-probabilistic.

Lines 140 to 161: In studies to investigate the effect of a variable on an outcome, it is best practice to keep every other variable constant (as much as is within the power of the investigator). The authors used only the Q fever Indirect ELISA by IDvet for livestock sample testing in Chad but used the same test only for equids in Switzerland. A different kit was used for ruminants. The authors were silent about this noteworthy discrepancy in their comparisons and discussions. Would the authors mind sharing why this does not influence their findings? Considering the study's design aimed to investigate the lab-to-lab or place-to-place differences in test results, is there a reason why the same tests were not used in the laboratories (also the RVF testing for Humans that used a different test in Europe)?

Lines 138, 148, 203, 206, 209, 218: It is unclear how the 10% mentioned in line 148 matches the sample numbers in 138. The final number of samples mentioned in lines 206 and 218 does not match the numbers in the last row of Table 1. Would the authors be implying that the total number of livestock samples used was 102 because 1 sample that was tested in the Q fever group could not be tested for RVF? If that is the message being conveyed, the authors may want to adjust line 206 to read, “91 human and 102 livestock samples were tested for either one OR both tests.” More text may be added in Figure 1 and other portions of the manuscript to make this clearer for the readers.

Line 197: the function exp(coefficients(model)) does not give an upper CI. Please revise lines 197 and 198

Minor concerns:

Lines 116-119: Please consider rewording your objective to be easier understood or piecing the objective into more than one sentence.

Figure 1: The authors may consider adding more details to the figure and title to let the readers grasp the full scope of what is being described without referring to the main manuscript text.

Reviewer #2: It is evident that the authors put a lot of effort into writing a well structured and detailed manuscript. The manuscript presents a clear purpose, informative literature review with clearly defined terms that can be easily understood by any reader. The research methods were also comprehensibly described and the results were well explained. The discussion has developed ideas and the authors provided relevant supporting data for their claims. There was a smooth transition between sections, giving a high level of cohesion and an overall high quality manuscript. 

The study is of utmost relevance in disease detection and control. The authors identified the need for considering demographics in disease detection which is novel and of great significance in field and laboratory studies, as well as in the development of diagnostic protocols. Hence, the manuscript is of high scientific value.

Reviewer #3: See attached file

Reviewer #4: The manuscript offers great insights on the challenge in disease diagnosis using methods in the context of poor resourced area. This paper could be of interest for local to regional levels readers. Manuscript topicality is excellent.

PLOS authors have the option to publish the peer review history of their article (what does this mean?). If published, this will include your full peer review and any attached files.

Reviewer #1: No

Reviewer #2: Yes: Dr. Ifeoluwapo Omolola Akanbi

Reviewer #3: No

Reviewer #4: No

Figure Files:

Data Requirements:

Reproducibility:

References

---

## [Decision Letter · Decision Letter 1]

30 Sep 2024

Dear DVM Hungerbühler,

We are pleased to inform you that your manuscript 'Diagnostic serology test comparison for Q fever and Rift Valley fever in humans and livestock from pastoral communities' has been provisionally accepted for publication in PLOS Neglected Tropical Diseases.

Best regards,

Philip Mshelbwala

Academic Editor

Elvina Viennet

Section Editor

Reviewer's Responses to Questions

**Key Review Criteria Required for Acceptance?**

**Methods**

-Are the objectives of the study clearly articulated with a clear testable hypothesis stated?

-Is the study design appropriate to address the stated objectives?

-Is the population clearly described and appropriate for the hypothesis being tested?

-Is the sample size sufficient to ensure adequate power to address the hypothesis being tested?

-Were correct statistical analysis used to support conclusions?

-Are there concerns about ethical or regulatory requirements being met?

Reviewer #1: Yes

Reviewer #2: The objectives of the study were clearly articulated and the study design well stated. The study population and sample size determination were properly described and appropriate for the tested hypothesis. The sample size was adequate and the statistical analysis were appropriate. Ethical approvals were duly obtained from appropriate institutions.

Reviewer #4: (No Response)

**Results**

-Does the analysis presented match the analysis plan?

-Are the results clearly and completely presented?

-Are the figures (Tables, Images) of sufficient quality for clarity?

Reviewer #1: Yes

Reviewer #2: The results were comprehensive and well presented. The contents of the tables were adequate and comprehensible. High quality data visualizations were used.

Reviewer #4: (No Response)

**Conclusions**

-Are the conclusions supported by the data presented?

-Are the limitations of analysis clearly described?

-Do the authors discuss how these data can be helpful to advance our understanding of the topic under study?

-Is public health relevance addressed?

Reviewer #1: Yes

Reviewer #2: The authors' conclusions were in line with the results of the study and their findings were extensively compared with other researchers' submissions.

Reviewer #4: (No Response)

**Editorial and Data Presentation Modifications?**

Reviewer #1: Accept

Reviewer #2: The manuscript has been improved upon after the first review and it is recommended for acceptance for publication.

Reviewer #4: (No Response)

**Summary and General Comments**

Reviewer #1: The authors have satisfactorily addressed my concerns

Reviewer #2: The study is very interesting and highly relevant. The authors adopted cutting edge methodology. Hence, the study is reproducible. The manuscript was well written, properly structured and would be easily understood by readers from a wide range of fields.

Reviewer #4: my concerns have been addressed and I recommend that the paper can be accepted.

PLOS authors have the option to publish the peer review history of their article (what does this mean?). If published, this will include your full peer review and any attached files.

Reviewer #1: No

Reviewer #2: **Yes: **Dr. Ifeoluwapo Akanbi

Reviewer #4: No

---

## [Editor Report · Acceptance letter]

8 Oct 2024

Dear DVM Hungerbühler,

We are delighted to inform you that your manuscript, "Diagnostic serology test comparison for Q fever and Rift Valley fever in humans and livestock from pastoral communities," has been formally accepted for publication in PLOS Neglected Tropical Diseases.

Best regards,

Shaden Kamhawi

co-Editor-in-Chief

Paul Brindley

co-Editor-in-Chief
